# The High-Performance Work System, Employee Voice, and Innovative Behavior: The Moderating Role of Psychological Safety

**DOI:** 10.3390/ijerph17041150

**Published:** 2020-02-12

**Authors:** Rentao Miao, Lu Lu, Yi Cao, Qing Du

**Affiliations:** 1School of Labor Economics, Capital University of Economics and Business, Beijing 100070, China; mrtmiao@hotmail.com (R.M.); 13141078566@163.com (L.L.); 2School of Psychological and Cognitive Sciences, Peking University, Beijing 100871, China; 3School of Foreign Studies, Capital University of Economics and Business, Beijing 100070, China; 15823922953@163.com

**Keywords:** high-performance work system, employee innovative behavior, voice behavior, psychological safety

## Abstract

In this study, we examined the associations of the high-performance work system (HPWS) with employee innovative behavior, and tested a theoretical model in which these associations were mediated by employee voice (promotive and prohibitive voice) and moderated by psychological safety. Matched data were collected from 46 HR (Human Resource) managers and 374 full-time employees from 46 companies in China with multi-source and time-lagged techniques. We found that the HPWS is associated with employee behavior. Both the promotive voice and prohibitive voice partially mediate the relationship between HPWS and employee innovative behavior. Psychological safety moderates the relationship between HPWS and the promotive voice. However, psychological safety does not moderate the relationship between HPWS and the prohibitive voice. Furthermore, psychological safety moderates the mediation effect of the promotive voice between HPWS and employee innovative behavior. We discuss the theoretical and practical implications of these findings.

## 1. Introduction

In China’s economy, innovation has become the primary driving force for development. Under the innovation-driven development strategy, organizational innovation, as a new engine of economic development, is playing an increasingly important role, and employee innovative behavior is key for enterprises to obtain sustainable competitive advantages. Employee innovative behavior responds to the constantly changing environment through reengineering and reinventing processes and methods. Therefore, employee innovative behavior, as a performance variable, has attracted growing attention from academic and practical circles [1], and is the fundamental guarantee of organizational innovation [2]. Research on employee innovative behavior continues to emerge. Extant research has suggested that it is often a result of distinctive individual, collective, or organizational features. Individual features mainly involve work characteristics [2], individual ability [3], cognition [4,5], goals [6], and social relations [7]. Collective-level features mainly involve leadership [8,9] and the leader–member relationship [4]. Organizational variables mainly include organizational climate [10,11], organizational culture [12], and the human resource system [13]. Previous studies on employee innovative behavior started from individual variables and paid more attention to team variables, especially leadership, because the individual’s initiative and creativity as well as exemplary and charismatic leadership play an autonomous and guiding role in inspiring innovative work behaviors in employees. However, it is undeniable that in the current fierce competition for talent, individuals with strong “subjective initiative” and leaders with high “autonomous mobility or selectivity” are difficult for organizations to control and restrain, leading to uncertainty and lack of sustainability in the organization. On the contrary, organizational rules, systems, and climate can be controlled by the organization. Thus these rules and systems could, on an overall and long-term basis and at the policy level, cultivate, guide, and motivate employee innovative behavior. Recent studies have also begun to focus on variables at the organizational level (such as organizational climate) to explore how to improve employee innovative behavior. The high-performance work system (strict recruitment, extensive training, employee authorization, salary management, result-based assessment, information sharing and employee competitive flow and discipline management), as an indispensable, stable, and sustainable institutional arrangement within organizations, has long been considered to be able to effectively stimulate employee behaviors [14]; however, the impact on employee innovative behavior has been neglected [15]. The term “high-performance work system” (HPWS) refers to a series of different but interrelated (internal fit and exterior synergy) human resource practices that affect employee knowledge, skills, abilities, and attitudes and ultimately improve employee and organizational performance. Therefore, it is necessary to study whether the organization can stimulate employee innovative behavior through the establishment of a system, that is, perfect human resource management practice.

Recently, the employee voice has been used to reveal the formation mechanism of organizational innovation [16]. The term refers to positive extra-role interpersonal communication behavior which expresses constructive opinions about workplace issues [17]. It is not only a concrete manifestation of individual participation in organizational decision-making, but also an important form of employee contribution to the organization. It not only helps to improve organizational innovation [16], organizational performance [18], and team performance [19], but also contributes to the improvement of employee job engagement [20], performance [21], and creativity [21,22]. It is recognized by Guo [23] as an important means and approach to realize employee, team, and organizational innovation. The employee voice may play an important role in revealing the shaping of formation mechanism of employee innovation. The strategic characteristic of the HPWS is also to encourage the employee voice [24]. The employee voice fully reflects employees’ goals with respect to further development of the enterprise; furthermore, employees need appropriate innovation motivation and the opportunity to use employee voice behavior. Self-determination theory [25] can be used to reveal this complex relationship. According to this theory, motivation is a necessary precondition for employee behavior. The coexistence of controlled-oriented practice and committed-oriented practice in the HPWS within China’s management context [14] increases the complexity of systemic impacts, has multiple effects on employees, and stimulates employees to develop motivations and to generate employee voice behaviors (employee promotive voice and prohibitive voice). These complex effects will further have an impact on employee innovative behavior. Therefore, employee voice may play an important role in revealing the formation mechanism of employee innovative behavior, and some scholars believe that it is necessary to explore its role in the mediation mechanism of HPWS [16]. So far, no related research has been found. This study will explore the mechanism of HPWS on employee innovative behavior from the perspective of voice behavior.

In addition, we need to acknowledge that the implementation of high-performance work systems does not necessarily bring about positive behaviors, and situational factors such as organizational culture and organizational context will interfere with the implementation of human resource systems, which will promote or inhibit the production of individual behaviors. Therefore, the generation of employee innovative behavior may be affected by organizational context [26]. For enterprises or employees, although there are potential failures or interpersonal risks that cannot be avoided in innovation, it has become a consensus that enterprises must innovate in order to maintain competitive advantage; however, whether employees choose to innovate or not is influenced by the organizational atmosphere [27]. As a group, when building a pluralistic and secure atmosphere, psychological safety determines the openness of information and the tolerance concerned with the decision process in the environment within an organization, and affects the smooth implementation of high-performance work systems. Further, it may affect the judgment of employees with respect to whether or not to adopt the innovative behaviors. In other words, the absence of psychological safety leads to a low degree of employee cooperation and organizational harmony, making it more difficult for an organization to implement measures such as encouraging information sharing and empowering employees to participate. Meanwhile, it will affect the fair assessment of the results, which is not conducive to employee innovative behavior. Therefore, psychological safety may be an important factor affecting the influence of HPWS on employee innovative behavior.

In summary, our study aims to explore which mechanisms and organizational contexts an organization can produce to encourage employee innovation. Based on the management situation of Chinese enterprises, employee voice (such as the promotive voice and prohibitive voice) is taken as the mediating variable between the high-performance work system and employee innovative behavior to test its mechanisms. Furthermore, the boundary of psychological safety with respect to the high-performance work system and employee voice was tested. The examination of the moderated mediation model helps to reveal the mechanism and boundary conditions of the influence of high-performance work systems on employee innovative behavior, which is of great significance to the high-performance work system, employee voice behavior, and innovative behavior.

## 2. Theoretical Background and Hypotheses

### 2.1. The High-Performance Work System and Employee Innovative Behavior

As the latest form of strategic human resource management, the high-performance work system has been widely studied by the academic circle. Although there is no consensus among academia on the content and assessment basis of the high-performance work system, related studies based on the situation of China are relatively inclusive, dealing with the applications characterized by commitment-oriented practices (western management) including strict recruitment, comprehensive training, employee involvement, information sharing, and incentive pay [28], as well as the applications characterized by control-oriented practices (China’s management situation) including results-based assessment, internal competitive flow, and discipline management [14]. Considering that China is in a period of profound transformation, the high-performance work system that gives consideration to commitment-oriented and control-oriented practices can more truly reflect the situation of enterprise human resource management [29].

The process of generating, adopting, and implementing new ideas is known as innovative behavior (IB). It includes multiple stages, such as identifying problems, stimulating ideas, proposing solutions, seeking the supporters of solution, implementing solutions, expanding production scale, and finally institutionalizing it [30,31], and covers all the discontinuous activities from idea generation to promotion and implementation of the idea [32]. Thus, it can be seen that the innovative work behavior can achieve the final output with clear application components and research value [33].

In addition, compared with single practice, the high-performance work system is more conducive to the stimulation of employee behavior [14]. Can it also effectively stimulate employee innovative behavior? The answer is yes. First of all, the high-performance work system can provide more professional training with higher quality, and individual learning theory [34] argues that “the knowledge update can maximize learning”. Therefore, the expansion of knowledge and the improvement of skills can promote individuals to perceive, think about, and solve problems from multiple perspectives. At the same time, employees are encouraged to share knowledge and information, so as to encourage employees to make innovate behaviors which is conductive to the process reengineering in the workplace and work redesign. While empowerment gives employees the right to make decisions and ownership, when employees perceive more job autonomy, they will think the enterprise attaches great importance to their work and is willing to provide help, and their confidence, beliefs, and optimism will be improved. Such emotions will improve the motivation and ability of employees to adopt new ideas to solve problems creatively and help employees to put new ideas into practice [35]. Incentive compensation can not only play an external incentive role through realizing individual expectations, but also plays an internal incentive role through the strengthening of the obligation to generate innovative behavior. Existing studies have also shown that it can effectively promote employee innovative behavior [2]. In addition, outcome-based assessment and competitive flow are generally regarded as challenge stressors rather than hindrance stressors by individuals, and the principles of competitiveness concerned provide opportunities for individuals to continuously learn, develop themselves, and achieve career success, thus facilitating the emergence of employee innovative behavior [10]. The optimal solution is to strike a balance between commitment-oriented and control-oriented practice in the Chinese management situation. Based on the job requirements–resources (JD-R) model [36], the high-performance work system will not only put forward work requirements through control-oriented practice, imposing challenging work pressure on employees, but will also provide work resources such as salary and development opportunities, while creating a supportive, participatory, and stimulating work environment through commitment-oriented practice [24], so as to promote the will and ability with which an individual commits to work tasks. Therefore, we predict that a high-performance work system that emphasizes systematicity and integrity is conducive to stimulating employee innovative behavior. Thus, we propose:

**Hypothesis** **1:***There is a positive relationship between the high-performance work system and employee innovative behavior*.

### 2.2. HPWS and Employee Voice

The “employee voice” refers to the expression of constructive opinions and ideas on issues related to the workplace, which is an active extra-role interpersonal communication behavior [17] and transformation-oriented organizational citizenship behavior [37]. Liang et al. [38] further divided it into the promotive voice and prohibitive voice. Promotive voice behavior (Promv) refers to employees proposing innovative solutions and suggestions with the motivation of cooperation in order to improve the state of organization, where the solutions and suggestions tend to focus on the ideal state that can be realized in the future. Since its focus is to improve the efficiency of the organization, it is easy to obtain extensive support from the organization. In contrast, prohibitive voice (Prohv) behavior refers to employees proposing preventive suggestions on issues that hinder the development of organizational norms so as to protect the organization from risks or potential crises, and thus avoiding the negative results of interpersonal or interest risks. 

The commitment-oriented practice of the HPWS is seen as a long-term investment in employees; according to the social exchange theory, employees are prone to generating a sense of obligation to return the organization [39,40], strengthening the sense of commitment and trust to the organization, and thus they are more inclined to put a forward promotive voice for the long-term development of the organization as a return for the organization. For example, team construction in the workplace as a flexible cooperative method of project will provide opportunities for employees to participate in management, greatly improve their work decision-making autonomy, and help employees and organization to generate clearer goals and expectations due to the formation of the team, so as to fully arouse the positive emotions of individual efforts for mutual expectations [1]. Thus, it encourages employees to put forward constructive advice for the interests of the organization. The high-performance work system also encourages the timely sharing of knowledge, skills, and information, attaches importance to and develops the employee voice behavior system, and creates an organizational environment of open communication and collision of ideas [24], so as to stimulate the willingness of individuals to express promotive voice. In addition, control-oriented practices of the HPWS, such as result-based appraisal system and the flow mechanism of talents, will provide employees challenging work pressure for the employees. However, in order to achieve excellent performance appraisal results or important posts in the organization, staff will put forward a promotive voice to attract the attention and recognition of high-level leadership. This allows employees to achieve career success and expectations. The empirical study also found that the HPWS implemented by the organization can positively influence the employee promotive voice through the mediating role of organizational support [27]. Therefore, we propose:

**Hypothesis** **2a:***There is a positive relationship between the high-performance work system and employee promotive voice*.

The prohibitive voice focuses on practices, behaviors, and events that deviate from organizational development. Compared with putting forward the promotive voice, individuals will face greater professional risks when putting forward the prohibitive voice. On the one hand, the commitment-oriented practice of the HPWS is conducive to the prohibitive voice, because the organization can expand and standardize the channels of the voice by establishing the employee voice system to demonstrate the organization’s preference towards employee voice behavior, so as to dispel employees’ doubts about the potential risks concerned with the prohibitive voice. At the same time, when conducting information-sharing communication, organizations often build full trust in their employees. When employees are aware of the poor production or financial deterioration of the organization, with adherence to organizational commitment and the principle of reciprocity, employees are also inclined to put forward the prohibitive voice to help the organization cope with difficulties. On the other hand, the control-oriented practice of the HPWS is also conducive to the generation of the prohibitive voice. When employees are faced with the pressure of the “elimination system”, they tend to point out obstacles and unreasonable phenomena in work. When they believe that the remaining rules and regulations or work procedures will not be conducive for them improving their work efficiency, and when facing the risk of the reduction of wages and the risk of unemployment, they are more likely to challenge the inefficient organizational system and put forward the prohibitive voice. Previous studies have shown that the HPWS positively affects employee prohibitive voice [27]. Therefore, we propose:

**Hypothesis** **2b:***There is a positive relationship between the high-performance work system and employee prohibitive voice*.

### 2.3. HPWS, Employee Voice, and Employee Innovative Behavior

The promotive voice behavior is one of the ways to help an organization innovate and successfully adapt to the dynamic competitive environment [38], and is also a key factor in the generation of innovative behavior [31]. Its focus is on the working standards that can be raised or the working processes that can be improved in the organization.

Employees can generate new ideas from different perspectives and provide constructive and reasonable suggestions for the organization, so as to gain the recognition and trust of the organization. Thus, employees are encouraged by the organization to bring up new ideas, and are more willing to seek supporters for their own suggestions or schemes in order to implement the schemes. Therefore employee innovative behavior is born.

Through conducting employee participation, authorization, and information sharing practices, high-performance work systems develop extensive cooperation between the employees, enrich the information sources for organization development, encourage employees to think from multiple perspectives, and provide an open atmosphere which is conducive in promoting innovation. With adherence to the principle of mutual benefit, employees will develop a sense of sound organizational commitment, trust, and identity [24], and thus propose a promotive voice for the interests of the organization. Meanwhile, the organization accepts and approves plans to increase the benefits of the organization or applicable suggestions to improve the organizational procedures. This is helpful to strengthening the willingness and confidence of employees to implement the plans, thus contributing to the generation of innovative work behavior. Therefore, we propose:

**Hypothesis** **3a:**The promotive voice mediates the relationship between HPWS and employee innovative behavior.

The prohibitive voice starts with the issues related to expectations and the phenomena which can be improved [41]. Employees identify potential problems which the organization is facing through reflection from multiple perspectives, and use the prohibitive voice to challenge the situation of the organization, through for example giving advice on inefficient systems, procedures, and unreasonable phenomena in the workplace. Firstly, due to the example and demonstration of voice behavior, the individual’s prohibitive voice behavior will encourage others to put forward more prohibitive voice behavior. Secondly, the leading role of voice behavior will push other employees to put forward more effective solutions to problems existing in the organization, and they will seek supporters for their voice solutions so as to promote the implementation of the solutions and promote the emergence of innovative behaviors.

Based on the social exchange theory, organizations can exchange the implementation of positive and beneficial human resource practices for employees’ reciprocal behaviors [42]. Firstly, the organization will create necessary conditions and supportive environments, such as voice channels and methods, for the prohibitive voice of employees through the implementation of the high-performance work system. On the one hand, individuals with loyal personality traits are willing to point out the existing problems of the organization due to the organizational commitments and trust brought about by the commitment-oriented practice of the high-performance work system. On the other hand, after considering and weighing personal risk and the risk of survival concerned with control-oriented practice, individuals with a low sense of loyalty also tend to put forward a prohibitive voice to improve their work situation. Such a prohibitive voice is conducive for the organization to identify and solve the potential risks and provides the direction of the long-term development for the organization. Furthermore, innovation arises along with the improvement of organization. In short, the implementation of high-performance work systems will not only affects employee prohibitive voice, but also stimulates innovation behavior under the effect of prohibitive voice. Therefore, we propose:

**Hypothesis** **3b:***The prohibitive voice mediates the relationship between HPWS and employee innovative behavior*.

### 2.4. The Moderating Role of Psychological Safety

Environmental factors have highly complex effects on individual innovation behavior and are key to a systematic and dynamic understanding of employee innovative behavior. Innovative behaviors challenge the development status of organization, inducing risks, uncertainty, and potential failure. As such, employees need to build trust and a sense of safety with the organization as a precondition and guarantee of innovation behavior [43]. In addition, as a kind of challenging extra-role behavior with high risks, employee voice behavior might have negative influence on employee interpersonal relationships or on their career, etc. In terms of organizational atmosphere and employee beliefs, “psychological safety” (PS) refers to a common belief that individuals respect and trust each other, and it emphasizes beliefs of safety in interpersonal adventure and risk-taking [44]. Employees should not be embarrassed, need not worry about being misunderstood or punished when they speak, and do not worry about interpersonal risks. Therefore, psychological safety can promote the desire and the generation of employee voice behavior to some extent and affect the decisions and judgement of employee voice behavior.

Organizations with greater psychological safety tend to be tolerant and encourage their employees to express immature ideas, and will understand, respect, praise, and protect employees, so as to lower the cost of the material and psychological cost of the staff voice, building up common goals of interest between employees and the organization. As a result, employees believe that the implementation of the high-performance work system is designed to encourage collaboration, information sharing, and helping each other. The organization allows free expression and is also willing to accept different opinions, and will show concern, attention, and encouragement, so as to meet the safety and utility demands of employees. In this situation, the probability of negative consequences is significantly reduced during individual pre-assessment, which increases the willingness of the employees to use promotive voice behavior and reduces their concerns about prohibitive voice behavior. At that time, even if there is a deviation in implementation of the HPWS, since employees hold safety belief built with psychological safety, employees will be endowed with sense of identity and sense of mission. Thus, they tend to regard themselves as an indispensable part of the organization and actively use voice behaviors for the development of the organization.

In an organization lacking psychological security, employees believe that the organization only aims to maximize its own interests and is only concerned about its own interests. Furthermore, the actual organizational atmosphere runs counter to the original intention of the construction of the organization. 

Therefore, the high-performance work system implemented by the organization will place emphasis on control-oriented practice and undervalue commitment-oriented practice, which leads to misunderstandings and confusion in terms of employees’ understanding of organizational systems, causing individuals in the workplace to focus more on competing with each rather than on sharing information. The over-protection of this information further reduces the trust among employees and even causes mutual suspicion, which leads to a decline in employee satisfaction and organizational harmony [45]. According to the requirements of humbleness and modesty, the idea that “harmony is expensive”, and the concept that one should “shoot the bird which takes the lead”, individuals are required to follow rules in their work and try their best to avoid unconventional behaviors. Employee voice behavior may cause internal conflicts and harm the interests of stakeholders. Therefore, compared with the long-term benefits of the organization, the impending losses brought on individuals caused by voice behavior will make employees realize that the cost of voice behavior is too high, inducing direct damage to their own interest. Furthermore, the “face consciousness” of the Chinese means that individuals avoid any words and deeds that may have a negative impact on themselves. Due to the lack of belief in psychological safety, employees are reluctant to use a promotive voice; besides, they will not be willing to use a prohibitive voice which comes with high interpersonal risks. Therefore we propose:

**Hypothesis** **4a:***Psychological safety moderates the relationship between the HPWS and employee promotive voice in such a way that when psychological safety is high the relationship between HPWS and employee promotive voice will be stronger*.

**Hypothesis** **4b:***Psychological safety moderates the relationship between HPWS and employee prohibitive voice in such a way that when psychological safety is high the relationship between HPWS and employee prohibitive voice is stronger*.

Thus, this study a moderated mediation model is formed. Specifically, the employee promotive voice and employee prohibitive voice mediate the impact of high-performance work system on employee innovative behavior, and the influence of the mediator depends on employee’s perception of psychological safety. Employees can have a higher level of psychological safety because high-performance work systems have a greater effect on facilitating employee promotive voice and employee prohibitive voice. Thus, the employee promotive voice and the employee prohibitive voice more effectively convey the effect of high-performance work systems on innovative behavior. In contrast, employees can have a lower perception of psychological safety because high-performance work systems have a weaker effect on facilitating employee promotive voice and employee prohibitive voice. Thus, the effect of high-performance work systems on innovative behavior is less likely to be conveyed through employee promotive voice and employee prohibitive voice. Therefore, we propose:

**Hypothesis** **5a:***Psychological safety moderates the indirect effect of HPWS on employee innovative behavior through employee promotive voice, in such a way that when psychological safety is high the HPWS will have a stronger relationship with employee innovative behavior*.

**Hypothesis** **5b:***Psychological safety moderates the indirect effect of HPWS on employee innovative behavior through employee prohibitive voice, in such a way that when psychological safety is high the HPWS will have a stronger relationship with employee innovative behavior*.

The theoretical model of this study is depicted in Figure 1.

## 3. Methodology

### 3.1. Research Sample

We collected data from 56 technology-based enterprises involved in the areas of Internet, communication, biological pharmacy, information technology, computer technology and environmental protection, construction engineering, consulting, and equipment manufacturing, as well as other industries, in Beijing, Tianjin, Hangzhou, and Chengdu. In order to avoid homologous deviation, this questionnaire was used for two investigations at two different points in time. At time-point 1, the measured variables included the high-performance work system (HPWS), which was evaluated by human resource directors of the enterprises (partially filled by deputy). Psychological safety was evaluated by the basic staff of the enterprises. Two months later, the measured variables at time-point 2 involved employee voice and employee innovative behaviors, both of which were evaluated by employees. In the first survey, 560 questionnaires on employee and 56 questionnaires on human resource managers were sent out. The numbers of collected employee–manager matching questionnaires which were valid were respectively 487 and 53. In the second survey, the respondents who had given their contact information were surveyed again. In the questionnaires which were taken back, invalid and data missing questionnaires were removed, and the remaining valid questionnaires were from 374 employees from 46 enterprises. In the valid samples, males represented 44.2%, and the average age was 35.41 (SD = 7.183). Unmarried individuals represented 51.7% of the sample. In terms of education, 43% were of college-level and below, 39.5% were undergraduates, and 17.4% had a master’s and above. The average work period was 11.374 years (SD = 8.825). With respect to income level (CNY), 29.7% earned CNY 5000 yuan and below, 33.7% earned CNY 5001–8000 yuan, 19.2% earned CNY 8001–12,000 yuan, 6.4% earned CNY 12,000–15,000 yuan, and 11.0% earned over CNY 15,000 yuan.

### 3.2. Measures

The key variables of this research were studied using a Likert scale (1 = strongly disagree, 7 = strongly agree).

*HPWS.* We adopted the high-performance work systems scale of Miao et al. [46]; the scale had seven dimensions including 22 items such as strict recruitment, extensive training, employee authorization, salary management, result-based assessment, information sharing and employee competitive flow, and discipline management. Some examples are as follows: “The company encourages employees to share professional knowledge and skills”, and “Compared with competitors, the company has a total compensation system of a high level”, etc. The internal consistency coefficient of variable alpha was 0.914.

*Employee voice.* The employee voice behavior scale developed by Liang et al. [38] based on the Chinese context was adopted, including two dimensions of the promotive voice and prohibitive voice, using the 11 high-load items. Sample items include “I will take the initiative to offer rational proposal to help company to achieve its goal”, and “When something goes wrong in the company, I dare to point it out and do not fear offending others”, etc. The internal consistency coefficient of variable alpha was 0.962.

*Psychological safety.* We adopted the eight items from the psychological safety scale of Anderson and West [47]. Sample items are as follows: “I usually share information in the team, rather than taking it as my own”, and, “It’s easy for me to be understood and accepted by other colleagues”, etc. The internal consistency coefficient of variable alpha was 0.960.

*Employee innovative behavior.* We adopted the eight-item establishment of innovation behavior questionnaire from Zhan et al. [48]. Sample items are as follows: “I always look for opportunities to improve the working methods and work procedure”, and “I often try to adopt a new method to solve the problems encountered in the work”, etc. The internal consistency coefficient of variable alpha was 0.939.

For questionnaires, the scales of “HPWS” and “Employee innovative behavior” were in the Chinese versions, while the scales of “Employee voice” and “Psychological safety” were in the English versions. 

Our team members (an associate professor, a PhD student, and two master students) translated the English scale into Chinese version, and then two PhD students whose major was English translated Chinese version into the English version again. After comparison with the original English version to make sure it was correct, we started to measure. 

*Control variables.* In order to rule out other explanations, we selected control variables at the individual level as suggested by previous research on innovative work behaviors. All control variables were measured at Time1. In this study, employee gender was coded as 0 = male, 1 = female, education level (1 = specialist or below, 2 = bachelor’s degree, 3 = master’s and above), marital status (0 = unmarried, 1 = married), income level (1 = 5000 and below, 2 = 5001 ~ 8000, 3 = 8001 ~ 12,000, 4 = 12,001 ~ 15,000, 5 = 15,000 and above), age, and tenure were taken as the control variables.

## 4. Results

### 4.1. Analysis Results of the Reliability and Validity

We use AMOS 21.0 (IBM, Armonk, NY, USA) to make confirmatory factor analysis of the collected data and used the model comparison method to examine the discriminant validity of each scale evaluated by employees. As shown in Table 1, the four model factors have the highest imitation degree among all the models. This shows that the constructs involved in this study had good discriminant validity and indeed represented four different constructs.

We used a multi-source, multi-layer, and longitudinal research design, but there were still four variables evaluated by staff (psychological safety; promotive voice and prohibitive voice; innovative behaviors). As such, in this study Harman’s single cause test was performed. The results showed that the explanatory covariate of the first factor was 27.458%, which did not reach half of the total explanatory amount of 62.973%. Therefore, the research data do not suffer from the serious issue of common variance [49].

Table 2 shows the main variables of the mean, standard deviation, correlation coefficient, and reliability coefficient. The correlation coefficients of each variable were from 0.022 to 0.582, meaning correlation was reasonable. Five factors of the Cronbach alpha coefficient were from 0.925 to 0.952, which are values far greater than an acceptable level of 0.700, which indicated that the measuring project had high reliability.

### 4.2. Hypothesis Testing

We use software of HLM6.0 (hierarchical linear models, belong to Scientific Software International, Cambridge, MA, USA) to examine the hypothesis. Specifically, this study used the high-performance work system as the independent variable and employee’s innovative work behavior as the dependent variable to build the model. In order to test the mediating effects of promotive voice and prohibitive voice, we adopted Baron and Kenny’s [50] three-step method. As shown in Table 3, in the first step, model 1 included employees’ gender, age, education, tenure, marital status, and income level as control variables. Model 2 incorporated the HPWS into the model, and the results showed that HPWS had a significant effect on employee innovation behavior (*γ* = 0.445, *p* < 0.01), Hypothesis 1 is supported. In the second step, model 5 and model 7 showed that the HPWS significantly predicted promotive voice (*γ* = 0.404, *p* < 0.01) and prohibitive voice (*γ* = 0.425, *p* < 0.01), and thus Hypothesis 2a and 2b are supported. In the third step, model 3 incorporated promotive voice into the model, and the HPWS was significantly correlated with innovative behavior (*γ* = 0.189, *p* < 0.01). When model 3 was compared with model 2, 0.189 < 0.445, and the promotive voice was significantly correlated with innovative behavior (*γ* = 0.634, *p* < 0.01). Therefore, promotive voice played a partial mediating role on the relationship between the HPWS and innovative behavior. Hypothesis 3a is supported. In the same way, model 4 incorporated prohibitive voice into the model, and HPWS was significantly correlated with innovative behavior (*γ* = 0.224, *p* < 0.01). When model 4 was compared with the model 2, 0.224 < 0.445, and the prohibitive voice was significantly correlated with innovative behavior (*γ* = 0.519, *p* < 0.01). Therefore, the prohibitive voice plays a partial mediating role in the relationship between HPWS and innovative behavior. Hypothesis 3b is supported.

For the test of the moderating effect in Hypothesis 4a and Hypothesis 4b, the relevant variables were first centralized. As shown in model 6 and model 8, the interaction effect of HPWS with psychological safety on promotive voice was significant (*γ* = 0.181, *p* < 0.01), and Hypothesis 4a is supported. However, the interaction effect has no significant effect on prohibitive voice (*γ* = 0.075, *p* > 0.05), and Hypothesis 4b is not supported. Meanwhile, Hypothesis 8 was not tested. Further, in order to test whether the effecting mode of regulatory effect in Hypothesis 6a was consistent with that in the hypothesis, we used the Aiken and West [51] method to draw the regulatory effect diagram. As shown in Figure 2, the relationship between HPWS and promotive voice was stronger when the psychological safety was higher (simple slope = 0.912, *p* < 0.01) and became lower when psychological safety was lower (simple slope = 0.570, *p* < 0.01). Hypothesis 5a is supported.

We tested the moderating effect of psychological safety on the indirect relationship between HPWS, promotive voice, and employee innovative behavior. As shown in Table 4, model 2 incorporated psychological safety into the model. In model 3, psychological safety had a significant moderation effect on HPWS and innovative behavior (*γ* = 0.224, *p* < 0.01), and the moderation effect coefficient was significant, which meets the first step of the criteria made by Muller et al. [52]. As shown in model 6 in Table 3, the coefficient of the interaction term between HPWS and psychological safety was significant (*γ* = 0.181, *p* < 0.01), which meets the second step of the criteria. In model 4, the effect of employee promotive voice on the innovative behavior was significant (*γ* = 0.594, *p* < 0.01), but after bringing the variable into the equation, the original moderated effect did not significantly reduce (*γ* = 0.224, *p* < 0.01 into *γ* = 0.117, *p* < 0.01). The effect value decreased significantly, and thus psychological safety moderates the mediation effect of promotive voice on the relationship between HPWS and innovative behavior, and Hypothesis 5a is supported; that is, when employee psychological safety is higher, the HPWS has a stronger effect on employee innovative behavior through promotive advice, and when employee psychological safety is lower, HPWS has a weaker effect on employee innovative behavior through promotive advice (as shown in Figure 3).

## 5. Discussion

Although studies on the influence of high-performance human resource practices on organizational performance and employee behavior are abundant, there is little research on employee innovative behavior. At the same time, previous works mostly discussed the influencing factors or effects of voice behavior, but few studies introduced it into the mediating mechanism to perform related studies. This study explored the internal relationship between the high-performance work system on the organizational-level and innovative behavior at the individual-level, and explored the mediating role of employee voice, providing important inspiration for organizational managers to pay attention to and encourage employees to actively use voice behavior. In addition, this study examines the possibility of the function boundary of employee’s psychological safety between high-performance work systems implemented by the organization and employee innovative behavior.

### 5.1. Theoretical Implications

Recently, some studies have begun to explore the relationship between high-performance work system and individual innovative behavior [13,53] in order to help the organizations more accurately understand the high-performance human resource practice that affects individual innovative behavior, and to guide employees to put efforts into the increase of enterprise competitive advantage through building organizational environment. Based on the work requirements–resources model, we found that the HPWS, which combines both control-oriented practice and commitment-oriented practice, has a positive effect on promoting individual innovation behavior, which is different from the notion that control-based or control-oriented practice measures such as results-based performance appraisal and the forced ranking system are not conducive to innovation behavior. It has some inspirations for the re-understanding of the dilemma of “boundary difficulty” in management practice [54]. 

Secondly, we put the two kinds of employee voice into the mediating mechanism to test their role in the relationship between the high-performance work system at the organizational level and the employee behavior at the individual level. The study found that the high-performance work system is conducive to the generation of employee promotive voice and prohibitive voice and can further promote individual innovative behavior. Compared with the prohibitive voice, the promotive voice transmitted the effect of HPWS more greatly with respect to on individual innovative behavior, further enriching the studies on the function mechanism of high-performance work systems. At the same time, the research results show that although the promotive voice and prohibitive voice are the mediating variables between the high-performance work system and individual innovative behavior, they only partially mediate the relationship between the high-performance work system and individual innovative behavior. Therefore, in the future, we should continue to explore the mechanism of the high-performance work system from different theoretical perspectives.

In addition, we discussed the interaction between the high-performance work system and psychological safety and found that the influence of the high-performance work system on promotive voice depends on psychological safety, and its influence on prohibitive voice has nothing to do with psychological safety. The research conclusion thus partially supports the theoretical hypothesis. The impact of the high-performance work system on employee promotive voice increased with increasing psychological safety, but the impact on prohibitive voice did not improve significantly with increasing psychological safety. Further, combined with the mediating effects and the moderating effects, it is verified that the impact of the high-performance work system on employee innovative behaviors partly depends on psychological safety, and the research results partially support the hypothesis. On the one hand, influenced by the traditional collectivism in Chinese culture [55], the idea that “harmony is expensive” (mean thinking) [23], and traditional ideas of interpersonal relationships, as well as “face consciousness”, when psychological safety is ensured, individuals are willing to use the promotive voice, so as to promote the generation of innovative behaviors. This not only aids the career promotion of employees, but also promotes the mutual benefits between individuals and organizations. Psychological safety significantly regulates the establishment and development of such a win–win relationship [56]. On the contrary, the prohibitive voice proposed by employees is likely to bring high costs. It is not affected by psychological safety and may be more dependent on individual characteristics such as loyalty and integrity. The competition for social resources is intensified, and the employment situation is grim. After constantly understanding, learning, compromising, and adapting to the external environment, individuals choose to keep their nose to the ground, restrain their sharp edges, and focus on their own work instead of taking risks and challenging the organizational system. The analysis of this result still needs to be further discussed based on China’s current conditions.

### 5.2. Practical Implications

First of all, employee innovative behavior is an important driver of “mass innovation” and the organization’s acquisition of a competitive advantage. It is the core competency urgently needed by Chinese enterprises facing economic restructuring and upgrading of the industrial structure. Human resource practice containing commitment-oriented practice and control-oriented practice has a positive impact on employee innovative behavior. In the contemporary era when “Buddhist Culture” prevails, Chinese enterprises should not only pay attention to the incentive of competition generated by control-oriented practice, but also give consideration to the humanistic care brought about by commitment-oriented practice. It can not only avoid the negative effect of the “Buddhism system” on social progress, but also promote the effective realization of the humanistic management concept and the good coordination to gradually improve the dilemma of “boundary difficulty”, so as to realize the improvement of organizational competitive advantage imperceptibly.

Secondly, the high-performance work system partially influences individual innovative behavior through employee voice behavior, indicating that the promotive voice and prohibitive voice jointly play an important role on the relationship between the organization’s high-performance work system and individual innovative behavior. Therefore, in the process of implementation of the high-performance work system, the organization should pay attention to the construction of team, the improvement of the voice behavior system, the encouragement of information sharing, and promotion of the fairness of result-based appraisal. The organization should respond to and effectively use the employee voice, a rare resource, so as to further promote individual innovation and ensure the long-term development of the organization.

Finally, the organizational context of psychological safety plays a positive role in employee promotive voice and innovative behavior. The HPWS cannot ensure that the information can always be accurately delivered to individuals in the way the organization have expected. If errors occur in the implementation of the HPWS or the individualized understandings of the employees, then the organizational atmosphere and employee belief constructed by psychological safety can encourage employees to share information and express themselves, improve the effectiveness of their voice, and tolerate failure, which is conducive to the promotive voice and thus leads to innovative behaviors. Similarly, organizations can also prevent the inhibitory effect of the promotive voice owing to lack of psychological safety through the implementation of the high-performance work system. Therefore, the high-performance work system and psychological safety have a interactive and coordinating effect on promotive voice and employee innovative behavior.

### 5.3. Limitations and Future Research

There are still some deficiencies in this study. First, although some hypotheses have been verified by data, considering the coverage, type, and level of the sample, the applicability of its further promotion still needs to be verified and discussed. Second, psychological safety fails to regulate the influence of the HPWS on prohibitive voice, which could be caused by theoretical construction, data sources, or statistical analysis methods, and the reasons need to be further explored. In the future, it is necessary to employ other theories to further explore the mechanism and the function boundary of the effect of the high-performance work system on employee innovative behavior, and to build employee innovative behavior, the core competency of the organization, from diverse dimensions so as to make contributions to the development of the organization and social progress.

## 6. Conclusions

Our study found that the high-performance work system implemented by the organization has a significant positive impact on employee innovation behaviors. Both the promotive voice and prohibitive voice partially mediate the relationship between the high-performance work system and employee innovative behavior. Psychological safety moderates the relationship between the high-performance work system and the promotive voice. Furthermore, the psychological safety moderates the mediating role of promotive voice between the high-performance work system and employee innovation behavior.

## Figures and Tables

**Figure 1 ijerph-17-01150-f001:**
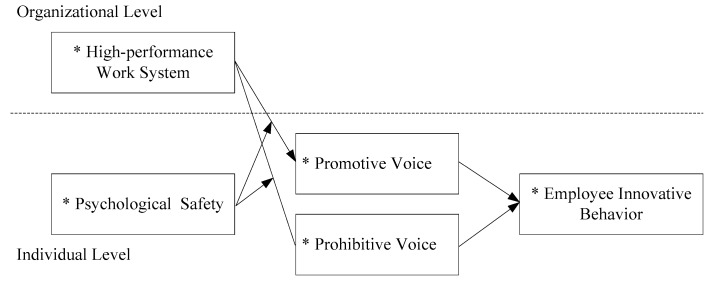
The conceptual multi-level model.

**Figure 2 ijerph-17-01150-f002:**
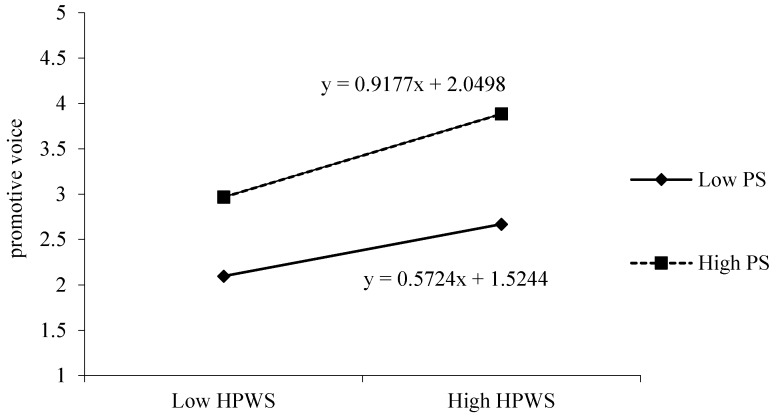
The interactive effect of HPWS and PS on promotive voice.

**Figure 3 ijerph-17-01150-f003:**
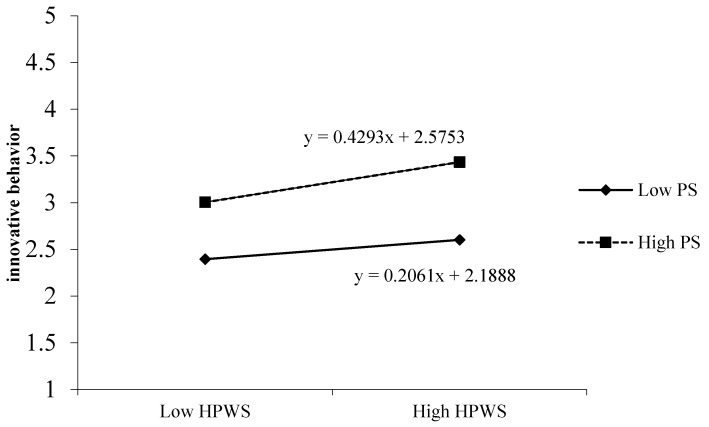
The interactive effect of HPWS and PS on innovative behavior.

**Table 1 ijerph-17-01150-t001:** Confirmatory factor analysis (*n* = 374).

Model	χ^2^	df	χ^2^/df	TLI	CFI	RMR	RMSEA
1. single factor model Promv + Prohv + PS + IB	4126.396	324	12.736	0.513	0.550	0.282	0.185
2. Two-factor model Promv + Prohv + PS	3546.735	323	10.981	0.585	0.619	0.272	0.171
3. Three-factor model Promv + Prohv	1756.045	321	5.471	0.814	0.830	0.138	0.114
4. Four-factor model	1062.321	318	3.341	0.903	0.912	0.111	0.083

*Note:* Promv: promotive voice; Prohv: prohibitive voice; PS: psychological safety; IB: innovative behavior; TLI indicates Tucker-Lewis index; CFI indicates comparative fit index; RMR indicates root of the mean square residual; RMSEA indicates root mean square error of approximation.

**Table 2 ijerph-17-01150-t002:** Mean, standard deviation, and correlations.

Variable	Mean	SD	1	2	3	4	5	α
1 HPWS	4.610	0.954						0.934
2 Promotive voice	4.611	1.269	0.111 *					0.952
3 Prohibitive voice	4.550	1.231	0.160 **	0.582 **				0.939
4 Psychological safety	5.206	1.088	0.529 **	0.022	0.141 **			0.935
5 Innovative behavior	4.932	1.055	0.132 *	0.522 **	0.413 **	0.110 *		0.925

*Note:**n* = 374; * *p* < 0.05, ** *p* < 0.01; HPWS = high-performance work system. The correlation coefficient in lower triangular matrix, the final as internal consistency coefficient alpha.

**Table 3 ijerph-17-01150-t003:** The test result of the hypothesis 1 to 5 (HLM).

Variable	Employee Innovative Behavior				Promotive Voice		Prohibitive Voice	
	M1	M2	M3	M4	M5	M6	M7	M8
*Individual Level*								
Gender	0.076	0.118 *	0.085 *	0.091 *	0.052	0.060	0.051	0.057
Age	0.084	−0.026	−0.022	0.064	−0.007	−0.001	−0.172	−0.156
Education	0.085	0.077	0.015	0.016	0.097	0.070	0.116 *	0.095
Openness to experience	−0.049	0.017	−0.063	−0.064	0.126	0.127	0.156	0.163
Marital status	0.002	0.117	0.021	0.012	0.151 *	0.172 **	0.201 **	0.220 **
Income level	0.048	−0.006	0.008	0.003	−0.022	−0.015	−0.018	−0.008
Promotive voice			0.634 **					
Prohibitive voice				0.519 **				
PS						0.209 **		0.252 **
*Organizational Level*								
HPWS		0.445 **	0.189 **	0.224 **	0.404 **	0.300 **	0.425 **	0.280 **
*Interaction*								
HPWS × PS						0.181 **		0.075
*σ^2^* within variation	0.485	0.471	0.420	0.406	0.322	0.429	0.353	0.417
*τ_00_* between variation	0.107	0.114	0.161	0.138	0.190	0.153	0.103	0.164
*Pseudo-R^2^*	0.042	0.091	0.119	0.112	0.069	0.148	0.083	0.134

*Note:* all are standardized coefficient; *n* = 374; M for the model; * *p* < 0.05, ** *p* < 0.01.

**Table 4 ijerph-17-01150-t004:** Test results of Hypothesis 7 (HLM).

Variable	Employee Innovative Behavior			
	M1	M2	M3	M4
*Individual Level*				
Gender	0.118 *	0.122 *	0.128 *	0.092 *
Age	−0.026	−0.006	−0.015	−0.014
Education	0.077	0.062	0.041	−0.001
Openness to experience	0.017	0.027	0.019	−0.056
Marital status	0.117	0.131 *	0.146 *	0.043
Income level	−0.006	0.004	0.005	0.014
Promotive voice				0.594 **
PS		0.236 **	0.306 **	0.182 **
*Organizational Level*				
HPWS	0.445 **	0.298 **	0.287 **	0.108 *
*Interaction*				
HPWS × PS			0.224 **	0.117 **
*σ^2^* within variation	0.471	0.438	0.415	0.450
*τ_00_* between variation	0.114	0.152	0.132	0.126
*Pseudo-R^2^*	0.091	0.113	0.144	0.139

*Note:* all are standardized coefficient; *n* = 374; M for the model; * *p* < 0.05, ** *p* < 0.01.

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
