# Peer review of "The High-Performance Work System, Employee Voice, and Innovative Behavior: The Moderating Role of Psychological Safety"

_ijerph, 2020, doi:10.3390/ijerph17041150_

Round 1
Reviewer 1 Report
I thank you for your efforts to research this timely topic. Your research effort has high relevance and potential for impact in terms of understanding of HPWS. Overall, this is an excellent paper.
I share the following suggestions for further enhancing and developing your manuscript. Please keep in mind while reading these suggestions that all of us scholars―and I am not even a slight exception here―have a lot of room to improve our research. The critical eyes of other researchers can sometimes elucidate these areas so we can refine our manuscripts. That is my intention here. At the same time, it is up to you to filter these comments and see if you find anything useful for future versions of your manuscript.
Introduction:
The authors used HPWS as an independent variable but throughout the introduction it is unclear that which specific practices are been considered for this study. As the author mentioned in line 383 that it has seven dimensions. Its better to write these dimensions where you define HPWS in the introduction. On page 2, line 55 “Recent studies have also begun to focus on organizational level” seems incomplete sentence. On page 2, line 61 “exerior synergy” is not clearly understandable, I think its “exterior synergy”. On page 2, line 87, the author says “employee voice may play an important role in revealing the formation mechanism ofemployee innovative behavior, and some scholars believe that it is necessary to explore its role in the mediation mechanism of HPWS (Rasheed, Shahzad, Conroy, Nadeem & Siddique, 2017)”. What basically the authors mean here? Please revise and make clear this sentence.
Literature:
Overall, the literature section is good and based on logical reasoning. In line 308, what the author means by “organizational system construction”
Methodology
This section provide enough details of participants and data collection procedure but no explanation is provided about the questionnaire. As the data collected from Chinese employees and the author didn’t explained either the questionnaire is distributed in English or Chinese language. If the Chinese version was used, explain the translation procedure.
Results
The results seem good and support the hypotheses.
Research conclusions and discussion
I suggest that the author should reorganize the “Research conclusions and discussion” part as follows. Firstly, author shall provide the discussion heading with a brief discussion on the study result. Next, the author shall provide theoretical implications followed by practical implications as a separate heading. This will make both these important portion more clearer to the audience. A conclusion shall be provided after “Limitation and future research” section.
General comments:
The sentences structure and other grammatical typos are frequently repeated throughout the manuscript. For example, sentence in line 170,171 etc. The author shall therefore, make sure to revise the manuscript with some native experts to eliminate such like mistakes. Citations and references are not accordingly with the journal format. The author needs to check the author guidelines on the journal website and correct it.
Reviewer 2 Report
Review report
Title: High-performance Work System, Employee Voice and Innovative Behavior: The Moderating Role of Psychological Safety.
The paper analyses the associations of high performance work system with employees innovation behavior with mediating role of employee voice and moderating role of psychological safety in the Chinese firm context. The paper is well written and the topic matches nicely the journal scope. The methodological approach is quite common in many organizational behavior literatures and it itself does not have much problems. Of particular, uncovering moderating role of Psychological safety on the indirect effect of HPWS on employee innovative behavior through employee voice have useful implications and contribution to extant OB literatures. However, there are a few minor areas which the authors must address carefully before this study could be published in IJERPH.
The structure of your introduction is probably not the best. As a result, the potential contribution of the paper is not apparent. The paper is not well motivated because the literature on organizational level factors affecting individual innovative behavior is not well-reviewed. Why is it important to examine the effects of HPWS in Chinese context rather than various other organizational variables (ex. Organizational culture, HRM practice) used by previous literatures? Development of hypothesis 3 should be enriched with proper citations of relevant previous literatures. Please provide the detailed information on overall analytical approach and process for hypotheses testing. Especially multilevel analysis by HLM you applied should be explained in more detailed manner. You would better include organizational size as an organizational level control variable in Table 4. You mentioned that in page 9 “Although we take the multi-source, multi-layer and longitudinal research design, but there are still four variables evaluated by staff, this study conducted Harman single cause. The results showed that the explanatory covariate of the first factor was 27.458%, which did not reach half of the total explanatory amount of 62.973%. Therefore, this research data has no serious problem of common variance.”.-Please also be clear with the explanation of “we take ~~longitudinal research design~”. I don’t agree with your claim that this research is a longitudinal research design. Please provide more proper references while treating common method bias problem and statistical test result you have done. The tables and figures should be thoroughly double-checked and revise the typo errors. For example Table 3 shows some Chinese letters in variable column. The references of thpaper should be thoroughly double-checked. All of missed and incorrect information of references should be amended. Discussion: in general, the discussion needs to be improved and tightened up linked to the results a bit more. What is it that we have learnt new from your effort? You may have to highlight your contribution a bit more clearly. The text of paper should be thoroughly proof-read and edited. All of the typographical errors, grammatical errors, odd expressions and lapses in English language should be double-checked and amended by the careful proofreading. (ex. Therefore, we predict that high performance work system that emphasizes Systematicity and integrity is conducive to stimulating employee innovative behavior.)

Round 2
Reviewer 1 Report
The author have done a good job and improved the article accordingly but still need a minor corrections in the revised manuscript.
In line 173 the word “Systematicity” write in the lower case as “systematicity”. Citations and references are not accordingly to the journal format.Also, a suggestion is here, in line 490 the heading “Research conclusions and discussion” could be replaced by only “Discussion” because conclusion heading is added at the end.
Author Response
Comments and Suggestions for Authors
The author have done a good job and improved the article accordingly but still need a minor correction in the revised manuscript.
In line 173 the word “Systematicity” write in the lower case as “systematicity”. Citations and references are not accordingly to the journal format.
Also, a suggestion is here, in line 490 the heading “Research conclusions and discussion” could be replaced by only “Discussion” because conclusion heading is added at the end.
Dear Professor
We are pleased to submit the second revision of the above manuscript. We ware gald that the evaluation we received on the first revision was positive—minor revision.
We would like to thank you for your careful consideration of the first revised version.
We hope that the new revised version and our handling of the comments will be deemed satisfactory for final acceptance. If , however, any additional work is needed we will be glad to work further.
Sincerely yours
The authors
RESPONSE TO THE COMMENTS OF THE REVIEWERS
The Reviewer notes:
"In line 173 the word “Systematicity” write in the lower case as “systematicity”. Citations and references are not accordingly to the journal format."
Authors' response:
We understand your point. In line with your suggestion, we revised it from "Systematicity" to "systematicity", and made some corrections for words and phrases.
In addition, we rearranged the references based on the journal format.
The Reviewer notes:
"Also, a suggestion is here, in line 490 the heading “Research conclusions and discussion” could be replaced by only “Discussion” because conclusion heading is added at the end."
Authors' response:
In line with your point, we replaced the heading "Research conclusions and discussion" with "Discussion".
